# Cardiac Connexin-43 Hemichannels and Pannexin1 Channels: Provocative Antiarrhythmic Targets

**DOI:** 10.3390/ijms22010260

**Published:** 2020-12-29

**Authors:** Katarina Andelova, Tamara Egan Benova, Barbara Szeiffova Bacova, Matus Sykora, Natalia Jorgelina Prado, Emiliano Raul Diez, Peter Hlivak, Narcis Tribulova

**Affiliations:** 1Centre of Experimental Medicine, Slovak Academy of Sciences, Institute for Heart Research, 84104 Bratislava, Slovakia; katarina.andelova@savba.sk (K.A.); tamara.benova@savba.sk (T.E.B.); barbara.bacova@savba.sk (B.S.B.); matus.sykora@savba.sk (M.S.); 2Instituto de Medicina y Biología Experimental de Cuyo, Consejo Nacional de Investigaciones Científicas y Técnicas, M5500 Mendoza, Argentina; nprado@mendoza-conicet.gob.ar (N.J.P.); diez.emiliano@fcm.uncu.edu.ar (E.R.D.); 3Department of Arrhythmias and Pacing, National Institute of Cardiovascular Diseases, Pod Krásnou Hôrkou 1, 83348 Bratislava, Slovakia; hlivakp@gmail.com

**Keywords:** cardiac arrhythmias, Cx43 hemi-channels, pannexin1 channels, mimetic peptides, cardio-protection

## Abstract

Cardiac connexin-43 (Cx43) creates gap junction channels (GJCs) at intercellular contacts and hemi-channels (HCs) at the peri-junctional plasma membrane and sarcolemmal caveolae/rafts compartments. GJCs are fundamental for the direct cardiac cell-to-cell transmission of electrical and molecular signals which ensures synchronous myocardial contraction. The HCs and structurally similar pannexin1 (Panx1) channels are active in stressful conditions. These channels are essential for paracrine and autocrine communication through the release of ions and signaling molecules to the extracellular environment, or for uptake from it. The HCs and Panx1 channel-opening profoundly affects intracellular ionic homeostasis and redox status and facilitates via purinergic signaling pro-inflammatory and pro-fibrotic processes. These conditions promote cardiac arrhythmogenesis due to the impairment of the GJCs and selective ion channel function. Crosstalk between GJCs and HCs/Panx1 channels could be crucial in the development of arrhythmogenic substrates, including fibrosis. Despite the knowledge gap in the regulation of these channels, current evidence indicates that HCs and Panx1 channel activation can enhance the risk of cardiac arrhythmias. It is extremely challenging to target HCs and Panx1 channels by inhibitory agents to hamper development of cardiac rhythm disorders. Progress in this field may contribute to novel therapeutic approaches for patients prone to develop atrial or ventricular fibrillation.

## 1. Introduction

Cardiac arrhythmias, such as atrial fibrillation (AF) and ventricular fibrillation (VF) are potentially life-threatening arrhythmias because of the high risk of stroke and sudden arrhythmic death [1]. Despite some differences in underlying mechanisms, these arrhythmias have been assumed to occur due to abnormalities in the electrical activity involved in impulse initiation and impulse propagation [1,2,3,4]. The former results from an enhanced automaticity of the cardiomyocytes (i.e., pacemaker-like activity) or triggered activity expressed as early or delayed after-depolarization. Conduction block promoting a re-entrant excitation (most likely due to an electrical uncoupling) results from dysfunction at the intercellular connexin (Cx) and ion channels, myocardial structural remodeling (hypertrophy and/or fibrosis, adiposity), and variations in the refractory periods [2,3].

There are three main factors involved in the development of severe arrhythmias: arrhythmogenic substrates, triggers, and modulating elements [3]. Arrhythmogenic substrates, i.e., myocardial structural remodeling and ion channel dysfunction in the setting of inflammation and oxidative stress facilitate the occurrence of both VF and AF [2,4,5,6,7,8]. Abnormal calcium handling and/or high intracellular calcium, acidosis, inflammation, and oxidative stress (mostly ischemia-related) may act as triggers [9], while alterations in autonomic tone are the modulating elements [10,11]. The triggered impulses often result from early or delayed after-depolarization due to the dysfunction of ions and Cx channels. The slowing or blocking of electrical signal propagation is a consequence of changes in passive and active membrane properties. These include alterations in membrane composition, impairment of gap junction Cx channels (GJCs)-mediated intercellular coupling, inhibition of Na^+^ current (excitability), and discontinuous tissue architecture, i.e., accumulation of extracellular matrix proteins and development of fibrosis [2,6,12,13,14].

Considering that cardiac arrhythmias result from a diverse and complex interplay of pathophysiological mechanisms, the approaches to preventing or treating arrhythmias need to be similarly complex, i.e., requiring a multitude of therapeutic options. Accordingly, progress in managing cardiac arrhythmias in the clinic includes autonomic tone manipulation, drugs and devices [10]. Despite of this complexity there is still a task to explore novel molecular targets for multi-targeted, reliable approaches aimed at preventing the development and/or recurrence of lethal VF as well as the growing incidence of AF.

In this context, the impact of direct intercellular communication mediated by cardiac GJCs is emphasized [4]. These nonselective intercellular channels ensure electrical and molecular signal propagation and contribute to the determination of myocardial conduction velocity [13,15,16]. The down-regulation and heterogenous cardiomyocyte distribution of GJCs along with cell-to-cell uncoupling is highly arrhythmogenic promoting triggered activity, conduction defects, dispersion of repolarization and refractoriness [12,15,16,17]. The over-expression of microRNA-1 results in the post-transcriptional repression of GJA1 genes for Cx and likely accounts for its arrhythmogenic potential [18]. The implication of micro-RNAs in human pathophysiology, in respect to arrhythmogenesis, has been recently suggested [19].

Very attractive are, therefore, recent investigations highlighting an important role of paracrine intercellular communication mediated by Cx hemichannels (HCs) as well as by pannexin channels in various tissues, including the cardiovascular system [20,21,22,23,24,25]. These channels are active in stressful conditions, and essential for paracrine signaling through the release of ions and molecules to the extracellular environment, and/or for their uptake. The opening of HCs and/or structurally similar pannexin1 (Panx1) channel profoundly affects intracellular ionic homeostasis, redox status, and facilitates pro-inflammatory and pro-fibrotic processes via purinergic signaling [22,25,26]. These conditions may promote myocardial arrhythmogenesis due to the impairment of the GJCs and selective ion channel function.

Despite the gap in knowledge concerning the regulation of HCs and Panx1 channels and their impact in the heart (see Section 2), evidence indicates that the activation of these channels can be pro-arrhythmic (see Section 3) and enhance the risk of cardiac arrhythmias (see Section 5). In this context, the mitochondrial HCs are of interest as well (see Section 6) in relationship with calcium homeostasis and reactive oxygen species (ROS) formation. This review aims to emphasize the role of cardiac HCs and pannexin1 channels as promising antiarrhythmic targets (outlined in Section 7).

## 2. A Brief Overview of Connexin Lifecycle and Topology of HCs and Pannexin Channels

Cx43 is the most expressed cardiac protein in the heart out of all the 20 rodents and 21 human connexin isoforms. As a typical integral membrane protein, Cx43 is synthesized in the rough endoplasmic reticulum and translocated into the Golgi compartment (Figure 1a) for folding and oligomerization into connexons. These are termed hemichannels (HCs) and they then follow the secretory pathway to reach their final destination at the plasma membrane [27,28,29,30,31,32,33].

The mechanism of how newly translated oligomerized proteins arrive at their proper location is still being explored [27,31,34].

Docking two HCs from neighboring cardiomyocytes into GJCs and assembly into gap junction plaques is quite prevalent at the intercalated discs, but rare at the lateral plasma membrane of cardiomyocytes in the healthy heart (Figure 1b,c). Connexin is characterized by four transmembrane segments with *α*-helical conformation and N- and C-terminal tails projecting into the cytoplasm. Two extracellular loops (E1 and E2), containing about 31 and 34 amino acids, respectively, are highly conserved and covalently connected by three invariant disulphide bonds. The extracellular domains E1 and E2 of the Cx43 have a critical function in the formation of the complete GJCs. Early phosphorylation and the presence of adhesion molecules appears essential prerequisites for HCs to GJCs assembly [27,28,29].

Some HCs in peri-junctional membranes (peri-nexus) do not undergo docking, and exert specific non-canonical functions beyond GJCs [25,34,35]. However, their mutual proximity implies integrated HCs and GJCs interaction and cooperation (Figure 1e,f). In addition, the peri-junctional membranes contain clusters of Na_v_ 1.5 channels that influence cardiac conduction [36], and these are combined with the large non-selective conductance channel constituted by pannexin1 (Panx1) [22,23,24,37,38]. There is also evidence for an ephaptic contribution to cardiac conduction which influences synchronization and timing of action potential firing [39,40,41]. The ephaptic conduction of electrical impulses occurs through transients in extracellular ions at the intercalated disc, and this prompts the activation of adjacent myocytes. This mechanism, however, appears to be localized in the peri-junctional region, and it implicates HCs and Panx1 channel activation.

Pannexins form channels at the plasma membrane surface that establish a pathway for communication between the cytosol of individual cells and their extracellular environment. Pannexins do not form intercellular GJCs, but only membrane channels. The pannexin family consists of 3 members, namely Panx1, Panx2, and Panx3, whereby Panx1 is the most expressed in the heart [42,43,44]. Panx1 was found to bind directly to actin, and this suggest that microfilaments act to stabilize Panx1 cell surface distribution or even participate in Panx1 delivery or internalization [38]. The current destination of HCs can also be determined by specific lipid rafts or membrane micro-domains associated with caveolae at the sarcolemma (Figure 1d). These are most important in compartmentalizing a variety of signaling activities [45]. Both HC and GJC are recycled via a post-endocytic event and this is consistent with rapid myocardial Cx43 turnover and =very short half-life of 1–2 h [46,47]. GJCs from the middle part of the gap junction plaque internalize to form a circular profile as “annular gap junctions” (Figure 1g). These undergo degradation by proteasomes and lysosomes [27,29,30,31,46]. However, the considerably rapid HCs/GJCs turnover is most important because it illustrates the dynamic nature of intercellular communication and Cx43 mediated signaling [25,27,35,37,48]. In contrast to the Cx43 protein, the Panx1 channels are composed of long-lived pannexin proteins which suggests a slower life cycle where internalized pannexins are degraded by lysosomes [38].

In addition to evidence that GJCs and HCs promote opposite responses in cellular signaling events, molecular studies indicate differences in GJCs’ and HCs’ dependence on intramolecular loop/tail interactions for their activity [35,49,50,51]. Monoclonal antibodies to the extracellular loop E2 enable HCs visualization of cell membranes without GJCs visualization because E2 epitope is not free in dodecameric GJCs [52,53]. These antibodies could aid HCs topology study through the changes in their myocardial distribution in response to acute and chronic heart injury associated with arrhythmias.

As illustrated in the diagram in Figure 1, GJCs, HCs and the Panx1 channels create a unique and dynamic signaling network in the heart and vascular system to ensure tissue homeostasis and proper function. This network coordinates myocardial contraction. It is also the most important in heart adaptation, and especially for the response to stressful or pathophysiological conditions by propagating both cell survival and apoptosis signals [20,22,25,37,38,46,51,54]. It is a fundamental and permanent task to control this network when trying to prevent or treat heart and life-threatening arrhythmias. Evidence suggests that pharmacological HCs and Panx1 channel blocking prevents their undesirable action following stress or injury, and also that the protective GJCs function could provide new treatment opportunities.

## 3. Implication of HCs and Panx1 Channels in Disturbances of Ionic and Redox Homeostasis as Well as Pro-Inflammatory and Pro-Fibrotic Signaling

Compared to GJCs, the HCs and Panx1 channels operate out of cardiomyocyte membrane gap junctions. Therefore, dominant myocardial HCs and Panx1 channel topology appears restricted to the lipid rafts/caveolae on lateral cardiomyocyte plasma membrane (Figure 1d). In addition, they also act at the peri-nexus [22], i.e., at the peri-gap junctional membrane (Figure 1e,f). HCs also perform the “non-canonical” functions of Cx43, beyond those of the GJCs and independent of gap-junction formation [25,55,56].

Findings from various non-cardiac tissues that might be relevant for cardiomyocytes indicate that HCs channels remain closed in physiological resting conditions, but change to open conformation as the extracellular Ca^2+^ concentration decreases or intracellular Ca^2+^ concentration increases, which open Panx1 channels as well [57,58,59,60,61] However, HCs and Panx1 channels do not have the exact same regulation. HCs tend to be activated by strong depolarization, while Panx1 channels activity may be induced at the resting membrane potential [62]. The pore diameter increases from 1.8 to 2.5 nm and this enables transmission of molecules up to 1 kD. HCs opening enables typical single-channel conductance of approximately 220 pS [63] and Panx1 channels have a large unitary conductance of around 500 pS. In addition, hydrophobic extracellular domains are also crucial in regulating Ca^2+^-dependent conformational changes [20].

The HCs and Panx1 channels can open following different extracellular stimuli. This is established in various non-cardiac tissues [57,58,59,64,65], but highly relevant to the heart. The HCs and Panx1 channels can therefore be activated by membrane depolarization, metabolic inhibition, and stresses including shear mechanical membrane forces and ionic and ischemic stress [24,38,57,62,66,67]. The HCs, and most likely also the Panx1 channels, are influenced by intra- and extra-cellular pH and phosphorylation and redox status [22,24,37,58,66,68,69,70]. All these events during heart disease development can activate both HCs and Panx1 channels and promote arrhythmias through bi-directional ion passage and small metabolic or signaling molecules below 1–2 kDa. Thus, HCs and/or Panx1 channels can mediate Na^+^, K^+^, Ca^2+^, cAMP, ATP, NAD+ influx as well as the transmission of glutamate, glutathione, prostaglandin-E2 and epoxyeicosatrienoic acid [24,35,71]. Panx-1 channels are known mostly as ATP release channels [60]. Intracellular localized Panx1 has also been proposed as a regulator of sarcoplasmic reticulum-based Ca^2+^ homeostasis [72].

HCs and Panx1 channel opening can profoundly affect intracellular ionic homeostasis, contribute to Na^+^ and Ca^2+^ loading and K^+^ loss, alter redox status, and facilitate pro-inflammatory and pro-fibrotic conditions [54,57,58,62,73]. These events impair GJCs function and produce myocardial electrical instability, thereby promoting malignant arrhythmias [4,32,66,74,75]. It was also previously noted that the activation of peri-junctional Panx1 channels [37] and Na_v_ 1.5 channels can influence cardiac conduction [36,39]. Abnormal Na_v_1.5 channel function along with autoantibodies against Cx43 and activation of Panx1 channels may underlie heritable arrhythmic syndromes [76,77,78].

HCs and Panx1 channel mediated ATP release is fundamental in the purinergic signaling which is so important in promoting inflammation in vascular function/dysfunction [62,65,66,79,80,81]. The inflammation is known to impair GJCs function and significantly contribute to arrhythmogenesis [74]. The Panx1 channels present at internal membranes may be implicated in K^+^ influx to mitochondria and intracellular Ca^2+^ leak/release from the endoplasmic reticulum [35,71,82]. Moreover, ATP binding to the P2Y receptors increases inositol 1,4,5-triphosphate which then releases Ca^2+^ from the ER and enhances ATP signaling to neighboring cells [24]. Disturbances in intracellular ionic homeostasis and Ca^2+^ overload promote triggered “post-depolarization” and GJCs uncoupling which render the heart prone to malignant arrhythmia development; and especially VF [83,84,85,86,87]. This triggered Cx activity and GJCs disordered expression and dysfunction are also implicated in AF development [85,88].

Cellular stress, including myocardial ischemia, enhances cardiomyocyte ATP release through the Panx1 channels that facilitate early fibroblast activation [66]. This suggests an early paracrine event leading to pro-fibrotic responses which could be involved in arrhythmogenic substrate development. In contrast, HCs inhibition by TAT-Gap19 alleviated tissue fibrosis and introduced the possibility of preventing arrhythmias [89]. Fibrosis precedes inflammation, and cardiac fibroblasts contribute to the inflammatory milieu through increased secretion of pro-inflammatory cytokines and chemokines released by the HCs and Panx1 channels [90,91]. Here, the Panx1 channels are interesting because of their implication in a variety of cellular responses [24,38], including apoptosis [92], inflammation [93], and innate immune processes [94,95]. Inflammation is also a prominent feature of arrhythmogenic cardiomyopathy [96]. Therefore, targeting inflammatory pathways could be an effective new mechanism-based therapy for familial non-ischemic heart muscle disease which causes sudden death in the young, and especially in athletes.

Figure 2 illustrates that functional HCs and Panx1 channels provide paracrine and autocrine communication pathways for ion and small molecule passage across the plasma membrane [22,37,38,66,97]. HCs and Panx1 channels may regulate many cellular processes but little information on their mechanisms is available. Extended information on HCs and Panx1 channel function from in vitro sources indicates their implication in diverse physiological and pathological responses relevant to the cardiovascular system [20,22]. However, the cardiac muscle remains ‘terra incognita’ and this provides challenges. This is especially important, in conjunction with rafts/caveolae related signal-transducing molecules, in the development of heart disease and life-threatening arrhythmias. The compartmentalized signal transduction appears to be an attractive area of research.

It is important that the caveolae and lipid rafts of cholesterol and sphingomyelin enriched membrane micro-domains are considered in HCs and Panx1 channels. These are all involved in the regulation of cell signaling pathways [58,59,98,99] and in GJCs homeostasis [100,101,102,103,104]. Caveolin-3 is the major caveolin isoform in cardiomyocytes, and this has been implicated in 17β-estradiol-elicited, rapid signaling to regulate Cx43 phosphorylation during ischemia [101]. Moreover, the caveolae/lipid rafts have been involved in reactive oxygen species production, redox signaling and K^+^, K_ATP_, Na^+^, and Ca^2+^ ion channel functions [105,106,107].

Cardiomyocytes and endothelial capillary cells are rich in caveolae which invaginate the plasma membrane, and this suggests the implication of these compartments in extensive HCs and Panx1 channel-mediated paracrine and autocrine signaling. The interaction of these channels and caveolins implies compartmentalized signaling, and this could be most important in the pathophysiological development of heart dysfunction and arrhythmias [45,103,104,108,109,110]. Investigation of this issue presents future challenges.

## 4. Impact of GJCs Abnormalities on Development of Malignant Arrhythmias

GJCs are part of multi-protein complexes and their function is tightly regulated [111]. They operate between cardiomyocytes in the gap junctions which are prevalent at the intercalated disc but rare at cardiomyocyte lateral plasma membrane. The fundamental importance of GJCs underlies electrical and metabolic coupling for rapid action potential transmission, and for signaling molecules between adjacent cardiomyocytes which synchronize myocardial contraction. Regular Cx43 myocardial expression and topology is essential for uniform anisotropic conduction [16,112,113]. This implies that aberrant Cx43 expression and/or its topology from cardiac or systemic disease and accompanying chronic or acute inflammation and oxidative stress have adverse impact on the heart.

Evidence clearly indicates that the down-regulation of Cx43, and its remodeling/lateralization through enhanced GJCs occurrence on cardiomyocyte lateral plasma membrane, are highly pro-arrhythmic and promote mechanical heart dysfunction [4,74,75,112,113,114,115]. In addition, acute GJCs uncoupling, mainly due to intracellular Ca^2+^ overload, acidosis, and ATP deficiency [116] slows the spatial spread of excitation [32,117,118]. This facilitates the occurrence of malignant arrhythmias, electro-mechanical dysfunction, and heart failure [4,75,83,116,119].

Myocardial architecture is important in development of cardiac arrhythmias, and GJCs topology facilitates the determination of patterns involved in activation and conduction velocity. Myocardial structural remodeling by hypertrophy, fibrosis, necrosis, or fatty deposits accompanied by both GJCs and ion-channel remodeling and dysfunction contribute to slower conduction, electrical instability, and de-synchronized contraction [75,113,117,118,120,121]. The poor prognosis of cardiovascular disease and chronic heart failure is related to increased susceptibility to lethal arrhythmias, such as VF [122,123]. Similarly, structural remodeling combined with altered GJCs and cellular electrophysiology aid the initiation, maintenance and progression of AF. This is the most common clinical arrhythmia and it jeopardizes life through high risk of stroke [2,85,88,124]. In contrast, the anti-arrhythmic cardiac phenotype is associated with preservation of Cx43 expression, phosphorylation, and distribution/topology [4,86,87,125,126,127].

## 5. Impact of HCs and Pannex1 Channels Activity on Development of Cardiac Arrhythmias

There is still a lack of information on HCs and Panx1 channel roles in cardiac muscle. Nevertheless, it has been shown that both HCs and Panx1 channels are the potential route for Na^+^ inflsux and K^+^ efflux during ischemia in isolated ventricular cardiomyocytes [39,40,62,128]. This can induce electrical disturbances [14,41]. The opening of HCs and Panx1 channels during ischemia has also contributed to re-perfusion injury following brief cardiomyocyte ischemia [66,91,129,130,131]. The activation of HCs and Pannex1 channels during ischemia/hypoxia, combined with other metabolic inhibition and post-ischemic reperfusion mechanisms may compromise cardiomyocyte ability to maintain ionic homeostasis. This is an essential step in promoting both arrhythmias and apoptosis [87,91,131,132,133,134].

Metabolic inhibition followed by pro-arrhythmogenic [Ca^2+^]_i_ and [Na^+^]_i_ overload in isolated cardiomyocytes was significantly reduced by halothane which decreases HCs conductance [135]. Importantly, HCs contribute to cytoplasmic Ca^2+^ oscillations by providing a bimodal Ca^2+^ dependent Ca^2+^ entry pathway [136]. Moreover, ischemia and/or hypoxia are associated with increased production of NO and S-nitrosylation [137,138] which is important in regulating HCs and Panx1 channel permeability [139,140]. Opening of connexin 43 hemichannels is increased by lowering intracellular redox potential [141]. The NADPH oxidase inhibitor apocynin prevented HCs activity by reducing nitroso-redox stress [142]. All these factors strongly suggest that HCs are implicated in arrhythmogenesis.

HCs and Panx1 channel-mediated purinergic signaling fulfills an important function in cardiac and vascular pathophysiology [22,79]. This includes ischemia, infarction, cardiomyopathy, hypertrophy, coronary artery disease, and ventricular and supraventricular arrhythmias [81,143,144]. The release of ATP via HCs or Panx1 channels in pathological conditions could induce post-depolarization triggered activity. This would generate ventricular tachycardia or VF [145,146]. Moreover, purinergic receptor activation by ATP induces ventricular tachycardia through membrane depolarization and Ca^2+^ homeostatic disorders [147]. Resultant sporadic Panx1 channel opening then triggered action potentials and promoted arrhythmogenic activity [23].

In addition, extracellular ATP induced shortened action potential duration in tissue preparations of atrial and ventricular myocardium and in the myocardial sleeves of pulmonary veins [148]. These events are known to precede AF [85,117]. The implication of ATP or adenosine in AF development, and its post-ablation recurrence, have been reported [149,150,151,152,153]. This AF was associated with a strong increase in atrial adenosine [154].

HCs and Panx1 channel mediation of purinergic signaling may also be involved in myocardial inflammation and proliferation [73,93,155,156], extracellular matrix alterations, and fibrotic structural remodeling [113]. These pro-arrhythmic factors are known to increase the risk of AF and VF and contribute to heart failure [75,85]. Ischemic or hypoxic settings lead to the opening of HCs and Panx1 channels in cardiomyocytes and to the release of ATP which activates fibroblasts to differentiate to myofibroblasts [66]. This HCs and Panx1 channel activation has been shown to be a key factor in the development of fibrosis in the injured myocardium linked with tissue repair [73,156]. Fibrosis precedes inflammation and cardiac fibroblasts contribute to the inflammatory milieu through the increased secretion of pro-inflammatory cytokines and chemokines released by HCs and Panx1 [90,93].

Experimental studies indicate that pro-fibrotic conditions occur in response to both infarction and cardiomyocyte injury following long-term systemic diseases accompanied by low-grade inflammation. These occur in hypertension, diabetes, dyslipidaemia, and in ageing [75,85]. Importantly, all interstitial, diffuse, patchy and compact patterns of myocardial fibrosis are obstacles to normal electric propagation. These promote conduction block and re-entrant excitation which are considered highly pro-arrhythmic. The severity of conduction disturbances depends on the amount and specific spatial distribution of excessive connective tissue within the myocardium [157].

In addition, non-myocytes in scar tissue can be electrically coupled to cardiomyocytes through GJCs, HCs, and Panx1 channels and contribute to electrotonic conduction across scar tissue [158,159]. Cx43 is increased in activated fibroblasts during fibrogenesis [91,160,161] and it enhances the possibility of electrically coupled cardiomyocytes and fibroblasts [160,161,162]. This hetero-cellular electrotonic coupling may facilitate conduction disturbances and arrhythmias [163]. Fibroblasts and myofibroblasts can alter cardiomyocyte excitation-contraction coupling through paracrine mechanisms [164], and this suggests the paracrine modulation of myocardial contractility and synchronized contraction. Fibrotic tissue, however, is an inefficient electrical signal conductor and cannot rhythmically contract, thereby jeopardizing critical myocardial functions [91]. This combination provides heterogeneous cardiac tissue, and this is a well-known predictor of arrhythmia risk.

Purinergic receptor P2 × 7 is activated by the extracellular ATP released by HCs and Panx1 channels, and this has been implicated in the cardiac fibrosis which results from most chronic inflammatory diseases [90,93,165]. P2X7 facilitates inflammatory NLRP3 assembly which leads to the secretion of pro-inflammatory factors which worsen both pro-arrhythmic phenotype and cardiac disease [166]. This exposure to inflammatory mediators then reduces GJCs mediated communication [167], where cytokines affect purinergic signaling by regulating the GJCs and HCs/Panx1 channels [21]. The HCs and Panx1 channels are mediators of inflammation and apoptosis [90,168,169], and they are therefore appropriate targets in modulating the adverse purinergic signaling which decreases GJCs’ function and promotes arrhythmias [170].

ATP released from cardiac fibroblasts through the HCs or Panx1 channels also activates pro-fibrotic P2Y2 receptors [171]. It appears that increased P2X1 receptor expression in the atria of patients suffering from dilated cardiomyopathy also increases the risk of AF [172]. The adverse cardiac structure remodeling which promotes arrhythmogenesis has been ameliorated by administration of the Cx43 mimetic Gap27 peptide in heart failure rats [173], and Gap26 is present in ischemic hearts [174,175,176].

Some studies [39,40], consider that opening the HCs and Panx1 channels localized at gap junction plaque edges contributes to the Cx-mediated adverse signaling following cardiomyocyte injury [91] through ATP efflux, pro-inflammatory molecules and excessive ions influx [130]. In contrast, the Gap 26/27 mimetics of the Cx43 extracellular loop inhibited injurious current from HCs and GJCs [176] and thus diminished myocardial injury [174]. Modest attenuation of the infarct size has been reported by Gap19 which is an intracellular loop mimetic that blocks HCs but not GJCs [63]. It appears that Gap19 and Gap 26/27 could be cardio-protectors from myocardial ischemia and reperfusion. The effects of these peptides on myocardial conduction in the injured or diseased heart and antiarrhythmic efficacy demand investigation. Moreover, their timing should be considered in therapeutic strategies intended to manipulate these proteins.

It is of interest that the Panx1 channel opening can be induced by virulent factors from Trypanosoma cruzi or HIV. These are associated with acute myocarditis through their enhanced ATP release and subsequent P2Y receptor activation. Pharmacological Panx1 channel blockade inhibits Panx1, ATP, and the P2Y2 pathway, thus inhibiting cardiac myocyte HIV replication and pathogen invasion [64,177].

Of note, GJCs and the HCs and Panx1 channels directly connect vascular cells, thereby coordinating their function and controlling vessel diameter and blood flow [178]. The HCs and Panx1 channels contribute to smooth muscle cell Ca^2+^ dynamics and contractility by facilitating Ca^2+^ entry, ATP release, and purinergic signaling [179]. Blocking the HCs and Panx1 channels can inhibit deleterious vasoconstriction. It is clear that ATP affects the heart and vasculature by providing energy, but via enhancing the purinergic signaling pathway promotes endothelial inflammation [180]. This latter function is currently important because of its therapeutic potential. This also includes the use of Cx43 mimetic peptides [181,182,183] which can protect against malignant arrhythmias. Although HCs and Panx1 channels present an attractive therapeutic target, their inhibition requires the development of more specific inhibitory agents.

## 6. Mitochondrial Putative HCs and their Impact on Cardiac Arrhythmias

In the context of cardiac HCs, attention should be paid to putative mitochondrial HCs that are formed upon the translocation of Cx43 from the cytosol to the mitochondria [82,184,185]. Translocation can be mediated by heat shock protein 90 and translocase of the outer mitochondria membrane [186] following PKCε activation with subsequent Cx43 phosphorylation [185]. An enhanced mito-Cx43 level has been induced by ischemic preconditioning, and this implies cardioprotective effects against ischemic heart injury [82,184,187,188,189], ischemia-reperfusion injury [20,186], and the regulation of apoptosis [190] and energy metabolism [122,191,192].

The implication of mitochondrial Cx43-mediated signaling in the attenuation of ischemia and/or reperfusion related electrolyte and redox misbalance and Ca^2+^ overload could undoubtedly be associated with protection against cardiac arrhythmias. The proposed modulation of the mPTP permeability transition pore by Cx43 signaling could have a fundamental role in this protection, but this issue requires further investigation.

Mitochondrial putative HCs, and perhaps the Panx1 channels illustrated in Figure 3 can modulate K^+^ influx by opening the ATP-dependent K^+^ and Panx1 channels [193]. Consequently, this would induce the mitochondrial depolarization associated with reduced mitochondrial ROS production and infarct size reduction [194]. In addition, the mitochondrial HCs channel, and most likely also the Panx1 channel, facilitates Ca^2+^ entry which affects matrix Ca^2+^ homeostasis and consequent mPTP. This was proven by blocking HCs with Gap27 [195,196]. The nitrosylation of mitochondrial Cx43 [137,197] and mitochondrial integrity protection [122] are implicated in this HCs mediated cardio-protection [198].

Both excessive intracellular ROS and Ca^2+^ overload are key factors that facilitate cardiac rhythm disorders [84,85,199,200,201]. It is plausible to suppose that mitochondrial HC and Panx1 channels could well be involved in determining the susceptibility of the heart to life-threatening arrhythmias. Data on mitochondrial HCs and Panx1 channel implication in heart disease, heart failure and malignant arrhythmias is lacking, and this invites challenging research.

It is interesting that putative mitochondrial HCs has been found solely in subsarcolemmal mitochondria and not in those located in the interfibrillar regions [187]. Transmission electron microscopy reveals that subsarcolemmal mitochondria are regularly located in the vicinity of long peripheral or lateral gap junctions, and even surrounded by gap junctions (Figure 4a–c). Moreover, subsarcolemmal mitochondria are often in the vicinity of sarcolemmal caveolae (Figure 4d) and mitochondria, gap junction and caveolae can all be closely located (Figure 4a,d,e). Sarcolemma-caveolae mediated signaling which targets subsarcolemmal mitochondria appears to have an important function in ischemic preconditioning [197]. In addition, Figure 4f–i show that even destroyed subsarcolemmal mitochondria are in the vicinity (enveloped) by gap junctions. This can be observed in pathological conditions accompanied by Ca^2+^ overload, such as ischemia, post-ischemic reperfusion, hypokalemia, and AF and VF (Tribulova, unpublished findings).

These data evoke a fundamental question. Does the demonstrated topography indicate mutual interaction (crosstalk) between cardiomyocyte GJCs and HCs/Panx1 channels present in the mitochondria and caveolae/rafts? This crosstalk could modulate GJCs and HCs/Panx1 channels function to support self-protecting myocardial signaling. This view is underpinned by the putative relationship between mitochondrial HCs and the mitochondrial mPTP [203]. The mPTP opens under stress, such as excessive ROS, inorganic phosphate, elevated matrix Ca^2+^ or reduced mitochondrial membrane potential. This latter leads to mitochondrial swelling and rupture which then facilitates cardiomyocyte injury and death. The mitochondrial HCs, and perhaps also the Panx1 channels, could regulate pathophysiological processes by inhibiting the mPTP opening [202].

Of interest, the apposition of plasma membrane caveolae and mitochondria enables caveolae-mitochondrial interaction, and this has been shown to regulate the adaptation to cellular stress by preserving the mitochondrial structure and function [204]. Moreover, caveolin transport into mitochondria and protection from ROS have already been reported [205,206]. It appears that the potential crosstalk between GJCs and HCs/Panx1 channels has important functions in cardiac pro-survival signaling and self-protection against life-threatening arrhythmias. This experimental paradigm, most likely relevant for the human heart, may reveal novel possibilities for targeted anti-arrhythmic approaches. The role of mitochondrial HCs in arrhythmia protection requires further elucidation. This equally applies to the Panx1 channels.

## 7. HCs and Panx1 Channels as Antiarrhythmic Targets

Post-translational modifications regulate the probability of GJCs channel opening, and conductance and selectivity. In addition to phosphorylation, post-translational Cx regulation includes ubiquitination, sumoylation, nitrosylation, hydroxylation, acetylation, methylation, and γ-carboxyglutamation [59,207,208]. There is less knowledge about the impact of post-translational modifications on HCs and Panx1 channel activity, and their effect on heart and vasculature also requires determination [22,38,209,210]. The GJCs, HCs, and Panx1 channels have specific conductance regulated by voltage, pH, and phosphorylation, therefore knowledge of channel-gating mechanisms is vital and demands exploration. That research remains most attractive because of the unprecedented putative HCs and Panx1 channel functions in arrhythmogenesis.

HCs channel-opening can be prevented by heptanol, octanol, carbenoxolone, α-glycyrrhetinic acid, flufenamic acid, and mefloquine but these agents also turn off GJCs [209]. Most pharmacological approaches do not distinguish between the GJCs and HCs or Panx1 channel functions in the heart, and new investigative molecular technology is required. There is increasing evidence that GJCs, HCs and Panx1 channels may also be counter-influenced by signaling molecules or intracellular events. Myocardial ischemia and reperfusion can lead to GJCs closure [116] and this is contrasted with HCs and Panx1 channel opening [20,22,178,211]. The oxidative stress accompanying pathophysiological events reduces GJCs [32] but activates HCs and Panx1 channels [20,22,60,212,213]. This could explain the transmission of injury current and the non-survival signaling associated with redox disorders and ionic leakage, as well as intracellular Ca^2+^ and Na^+^ overload and resultant cell death [128,129,132,134,135,211,214]. This clearly indicates that GJCs closure combined with HCs and Panx1 channel opening can facilitate malignant arrhythmias. It is also supported by the knowledge that elevated [Ca^2+^]_i_ triggers HCs and Panx1 channel-opening but inhibits GJCs opening. Moreover, Ca^2+^-calmodulin interaction with Cx43 is most important in this regulation [215,216].

Apart to their cardiomyocyte expression, the HCs and Panx1 channels are expressed in endothelial cells, fibroblasts, and leukocytes [22,60,159,217]. Recent evidence implicates extracellular ATP as an important inflammatory signaling molecule and mediator of multiple early inflammatory mechanisms, including neutrophil circulation. There is also a major ATP release pathway to the extracellular space through HCs and Panx1 channels in all the above-mentioned cell types. Cytokines released during inflammation [218] inhibit GJCs and reduce intercellular communication while enhancing HCs and Panx1 channel activity and increasing plasma membrane permeability [22,26,93,156,217,219,220,221].

Inflammatory liposaccharides and basic fibroblast growth factor cause GJCs inhibition while stimulating ATP release through glioma cell HCs [222]. These mechanisms may further promote inflammation and a resultant deteriorated disease state. Oxidative stress decreases GJCs mediated intercellular communication, but concurrently increases both channel opening and the number of surface astrocyte HCs [139,141,223]. This increase requires elevated [Ca^2+^]_i_ [69,223], and it suggests that Ca^2+^ overload may enhance the number of active HCs in cardiomyocytes which then contribute to GJCs dysfunction and increased heart vulnerability to arrhythmias.

Ample evidence indicates that GJCs, HCs, and Panx1 channels are counter-regulated. This was recently highlighted by differential growth factor action [224]. Gap26 and TAT-Gap19 divergently affect in vitro macrophage HCs in addition to astrocytes [225]. They also differentially alter inflammatory response outcomes in lethal in vivo sepsis [226]. Moreover, inhibition of HCs mediated ATP release by the flufenamic acid HCs blocker attenuates early inflammation [217].

Findings in inflammation and oxidative stress in non-cardiac tissues could also be relevant to the heart. ATP released to the vasculature through HCs and Panx1 channels provides the driving stimulus for early inflammatory response to implanted devices in heart disease. It is therefore challenging to explore HCs and Panx1 channel function in inflammation and oxidative stress which precede or accompany heart disease. These include atherosclerosis, dyslipidemia, hypertension, diabetes, and obesity, as well as heart failure and acute myocardial infarction, ischemia, and reperfusion. This stresses the fundamental impact of underlying intracellular redox imbalance on HCs and Panx1 channel function. It may also significantly contribute to the development of arrhythmogenic substrate and the occurrence of malignant arrhythmias. Comprehensive research is therefore required.

## 8. Concluding Remarks and Future Prospective

Cardiac connexin-43 and pannexins are transmembrane protein forming channels with diverse topology and functions. The GJCs are dodecameric pores at specific cardiomyocytes gap junctions. In contrast, the HCs (hexameric) and Panx1 (heptameric) channels are pores located in peri-junctional space, lipid rafts, and caveolae and inner mitochondrial membranes. This topological diversity concurs with multiple potential functions that are still under intensive research. There is apparently sophisticated crosstalk between GJCs and HCs/Panx1 channels, and this creates further challenges to explore. GJCs and the HCs and Panx1 channels are highly dynamic structures regulated by complex mechanisms, rather than just passive conduits. GJCs function can be impaired in pathophysiological conditions where the HCs and Panx1 channels thrive. Dysregulated Cx43 expression and topology combined with HCs and Panx1 channel activation impairs both GJCs and ion channel function and promotes life-threatening arrhythmias. Unfortunately, much less is known about activated HCs and Panx1 channel functions in this process than the well-established role of dysfunctional GJCs in arrhythmogenesis.

In conclusion, the available data strongly implicates cardiac HCs and Panx1 channel mediated paracrine signaling in pro-arrhythmic mechanisms and highlight further research. Further future research could also develop stable Cx43 mimetic peptides and agents. These could then confine their blocking actions to HCs and Panx1 channels without interfering in GJCs’ functionality, and thus avoid pro-arrhythmic consequences. It is important to inhibit HCs and Panx1-channel mediated adverse processes that facilitate the development of cardiac arrhythmia as well as to prevent/attenuate heart failure resulting from acute and chronic pathophysiological conditions. This paper adds valuable information for targeted cardiac HCs and Panx1 channel strategies that may protect people from life-threatening arrhythmias.

## Figures and Tables

**Figure 1 ijms-22-00260-f001:**
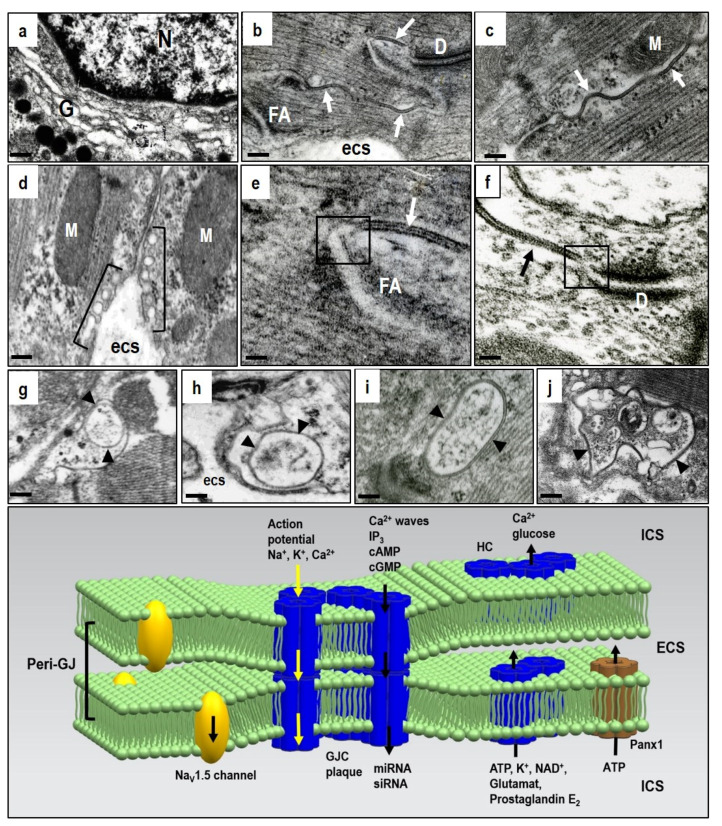
Electron microscopic images of the adult rat heart cardiomyocytes showing structures hosting Cx43. (**a**)—Golgi compartment in which Cx43 is oligomerized to HCs; (**b**–**d**)—post-Golgi HCs underwent targeted delivery to the particular membrane-subdomains, i.e., caveolae/rafts and gap junctions, close apposition of neighboring cardiomyocytes, which consist of paired HCs forming gap junction channels (GJCs). Gap junction plaques (arrows) are prevalent at the intercalated disc (**b**) and scarce at the lateral sides of the cardiomyocytes (**c**), while caveolae/rafts (brackets) are abundant at the lateral membranes (**d**). Peri-junctional plasma membranes (boxes in (**e**,**f**)) contain undocked HCs and Panx1 channels. The internalization of gap junctions into annular profiles (arrowheads in **g**–**j**) precedes degradation of HCs via proteasomal and lysosomal pathways. Abbreviations: G—Golgi compartment, N—nucleus, D—adhesive junction, desmosome, FA—fascia adherent junction, M—mitochondria, esc—extracellular space, scale bar—0.1 micrometer. The diagram demonstrates the putative topology of GJCs at the gap junction plaque, HCs and Panx1 and also the Na_V_1.5 channels at the peri-junctional membranes together with transmitted ions and molecules; ICS—intracellular space, ECS—extracellular space. Tribulova, unpublished representative electron micrographs.

**Figure 2 ijms-22-00260-f002:**
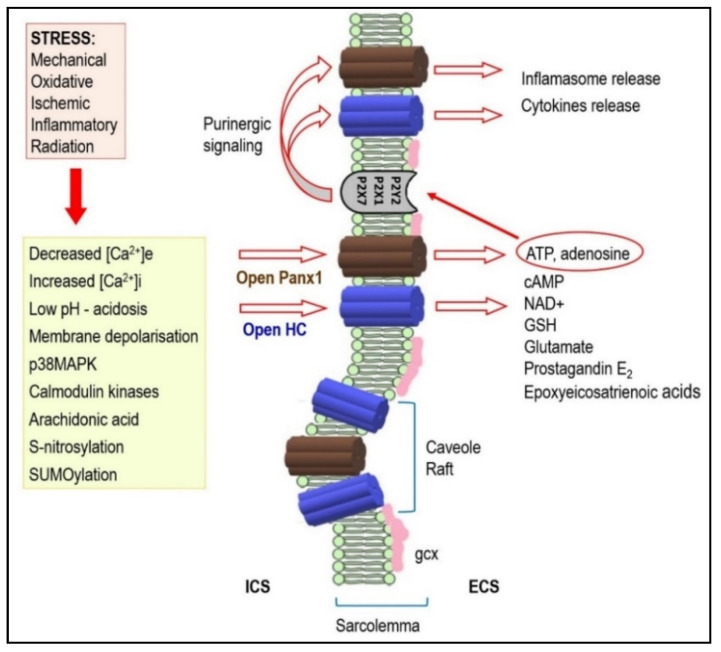
Under stressful conditions, several intracellular signals regulate hemi-channels (HCs) opening. Activated HCs enables release of signaling molecules to the extracellular environment. Therefore, it acts as a paracrine and autocrine communication pathway. HCs mediated ATP release is important in the purinergic signaling which is so important in cardiac pathophysiology. It may also be implicated in arrhythmogenesis (see text for details). Abbreviations: ICS—intracellular space, ECS—extracellular space, gcx—glycocalyx.

**Figure 3 ijms-22-00260-f003:**
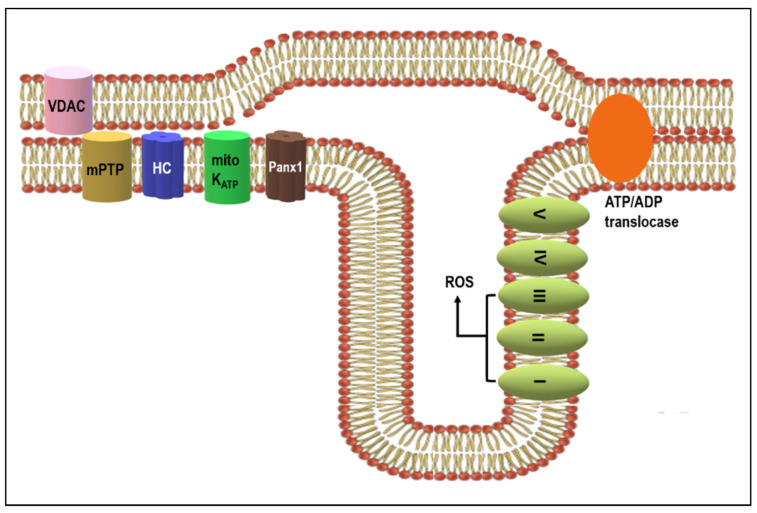
Diagram illustrating possible implication of putative mitochondrial HCs and Panx1 channels in cardioprotective signaling pathways promoting the opening of K_ATP_ channels and hampering the mitochondrial permeability transition pore (mPTP). This may result in the prevention of Ca^2+^ overload, excessive levels of ROS and decline in ATP production. For details, see text and references [187,202].

**Figure 4 ijms-22-00260-f004:**
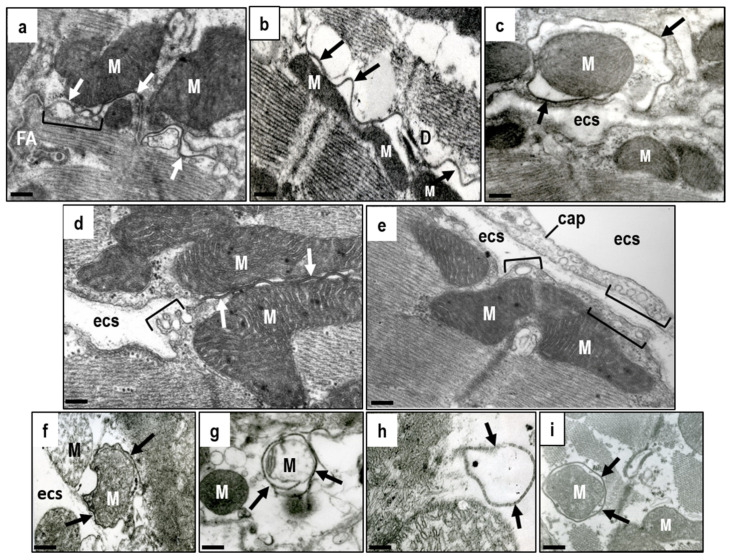
Electron microscopic images of rat heart cardiomyocytes illustrating the intimate relationship between GJCs located at gap junction plaques (arrows) and mitochondria (M) or caveolae/rafts (brackets). This implies possible signaling crosstalk between those entities. (**a**) Long gap junction (arrows) at the intercalated disc in the vicinity of subsarcolemmal mitochondria (M); (**b**,**c**)—Long gap junction on the lateral aspect of the cardiomyocyte touching or surrounding mitochondria; (**d**)—Long gap junction touching mitochondria in the vicinity of caveolae/rafts; (**e**)—Mitochondria in the vicinity of sarcolemmal caveolae/rafts and capillary rich in caveolae/rafts; (**f**–**i**)—Myocardial ischemia and Ca^2+^ overload result in cardiomyocyte injury and the loss of mitochondrial integrity, whereby there is a close relationship between gap junction profile and mitochondria even in those conditions. This suggests mutual GJCs, HCs and Panx1 channels communication. Abbreviations: ecs—extracellular space; FA—adhesive fascia adherent junction; D—adhesive junction-desmosome, cap—capillary, scale bar—0.1 micrometer. Tribulova, unpublished representative electron micrographs.

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
