# Peer review of "Cardiac Connexin-43 Hemichannels and Pannexin1 Channels: Provocative Antiarrhythmic Targets"

_ijms, 2020, doi:10.3390/ijms22010260_

Round 1

Reviewer 1 Report

I read article entitled ‘Cardiac connexin-43 hemichannels and pannexin1 channels: provocative antiarrhythmic targets' with great interest.

This review paper concerns important topic: potential clinical utility of connexin-43 hemichannels and pannexin1 channels in cardiac arrhythmogenesis.

After initial excitement on the topic as a clinician-scientist I did not find any useful and easy to get clinical knowledge from this paper. There is insufficient data claiming the clinical utility of these potential drug targets.

Few tables relating these potential targets to data from human studies (including those on the presence of arrhythmia – clinical studies) would improve the paper.

Why the authors think that these (hemi-)channels are more promising than other myocardial channels?

How currently used antiarrhythmic drugs affects these (hemi-)channels?

Moreover, the methodology of such a paper should be a systematic review and be performed in accordance to appropriate international guidelines (e.g. PRISMA guidelines).

Author Response

Dear reviewer,
we would like to thank you for your time to read our manuscript and we appreciate very much your critical clinical view. Unfortunately, we are very sorry that because the topic of this review is based on experimental data that challenge further investigation we could not response to your suggestions. However, we believe that this review is indeed challenging for further basic research as well as preclinical studies that may provide a new information relevant for treatment malignant arrhythmias in clinic.
Dear Editors and Authors, I read article entitled ‘Cardiac connexin-43 hemichannels and pannexin1 channels: provocative antiarrhythmic targets' with great interest. This review paper concerns important topic: potential clinical utility of connexin-43 hemichannels and pannexin1 channels in cardiac arrhythmogenesis.
After initial excitement on the topic as a clinician-scientist I did not find any useful and easy to get clinical knowledge from this paper. There is insufficient data claiming the clinical utility of these potential drug targets.
You are right that there is insufficient data supporting the idea that Cx43 hemichannels and pannexin channels might be an antiarrhythmic target. Therefore, we put in the title questioner. However, as presented in Section 3 and particularly in Section 5 in revised manuscript, it is clear that activation of these channels is pro-arrhythmic. In particular, their activity during inflammation and oxidative stress, which accompanies heart diseases or myocardial injury (in patients as well). To understand the link between cardiac arrhythmias and hemichannels we included new chapter 1.
Few tables relating these potential targets to data from human studies (including those on the presence of arrhythmia – clinical studies) would improve the paper.
Unfortunately, clinical studies are missing but we are optimistic for future. See please most recent editorial in European Heart Journal, 2020, 41, 4100–4102: The sooner, the better: anti-inflammation in acute myocardial infarction as well as associated paper of Bouabdallaoui N. in this issue.
Why the authors think that these (hemi-)channels are more promising than other myocardial channels?
Maybe because we are focusing on hemichannels in this review it makes impression that we eliminated another ions channels. However, we included them as well (see f.e. Na+,K+ and Ca+ channels in the text). We also noted that activation of hemichannels can impairs function of these selective ion channels.
How currently used antiarrhythmic drugs affects these (hemi-)channels?
This is very good question but no answer yet. However, we think that not only antiarrhythmic drugs (e.g. beta blockers) but also cardioprotective drugs (e.g. statins, ARB) may influence (inhibit) hemichannels. It should be, however, proved.
Moreover, the methodology of such a paper should be a systematic review and be performed in accordance to appropriate international guidelines (e.g. PRISMA guidelines).
We used PubMed data base and included most relevant and most recent studies as it is generally used when preparing basic research review.

Reviewer 2 Report

The present review relates to Cx43 hemichannels and pannexin1 channels in heart and their role in cardiac rhythm. More importantly they addressed their potential role in relation to gap junction channels. This review is clear and well written. It is also well illustrated and documented with numerous references. However, I have few comments to improve the manuscript.

Page 2 line 60 and page 3 lines 92-93, the authors refer to E1 and E2 without explaining what is it. They should describe the structure of the connexins in order to better understand why E1 and E2 are important.

Sections 2 and 3 deal with similar subjects. Namely, they both refer to HC, panx1 and ions (Na+, Ca2+) and ATP and the pro-arrhythmic nature of these channels. The authors should merge both sections and clarify this point.

Author Response

Comments and Suggestions for Authors
The present review relates to Cx43 hemichannels and pannexin1 channels in heart and their role in cardiac rhythm. More importantly they addressed their potential role in relation to gap junction channels. This review is clear and well written. It is also well illustrated and documented with numerous references. However, I have few comments to improve the manuscript.
Thank you very much for your time to read our review manuscript and for your comments and suggestions. We appreciate very much your positive view on our effort to discuss the topic. Thanks.
Page 2 line 60 and page 3 lines 92-93, the authors refer to E1 and E2 without explaining what is it. They should describe the structure of the connexins in order to better understand why E1 and E2 are important.
According to your suggestion, we have added following text highlighted in blue. Connexin is characterized by four transmembrane segments with α-helical conformation and N- and C-terminal tails projecting into the cytoplasm. Two extracellular loops (EL1 and EL2), containing about 31 and 34 amino acids respectively are highly conserved and covalently connected by three invariant disulphide bonds.
Sections 2 and 3 deal with similar subjects. Namely, they both refer to HC, panx1 and ions (Na+, Ca2+) and ATP and the pro-arrhythmic nature of these channels. The authors should merge both sections and clarify this point.
We agree with you that it seems that Section 2 is similar to Section 3. However, Section 2 provides general information dealing with HC and Panx1 channel function observed mostly in non-cardiac tissues. While Section 3 provides evidence about HC and Panx1 channel function observed dominantly in the heart tissues and in close association to arrhythmogenesis. Therefore, we would like to keep it both if you would agree.
Nevertheless, in revised version the title of Section 2 (now is 3) was changed for Implication of HC and Panx1 channels in disturbances of ionic and redox homeostasis as well as pro-inflammatory and pro-fibrotic signaling. The title of Section 3 (now is 5) was changed for Impact of HC and Panx1 channels activity on development of cardiac arrhythmias

Reviewer 3 Report

Major concerns

  • This review needs extensive editing work from the organization and English language point of views. For example, sub-paragraphs within each chapter need to be more focused and separated from each other’s. It reads more like a mismatch of information. There is a lack of flow, transitions, and logical progression.
  • Recent studies of Panx1 structure by cryoEM revealed a heptameric assembly, not hexameric (Nature Structural & Molecular Biology volume 27, pages373–381(2020); eLife 2020;9:e54670; Nature volume 584, pages646–651(2020)). Please correct line 459.
  • Chapter titles do not seem to always match their content or there seems to describe overlapping topics. For example, inflammation or fibrosis formation due to activation of the purinergic receptor through release of ATP is repeated in Chapter 2 and 3.
  • Figure 3 legend is misleading. Looks like a representation of the channels permeability rather than “illustrating possible implication of putative mitochondrial HC and Panx1 channels in cardioprotective signaling pathways”. May need to be revised to clearly convey your message.
  • Too many references seem to be reviews. References to actual research papers should be preferred.
  • Cx hemichannels and Pannexin channels do not have the exact same function and regulation. A note on their differences and not considering them as the same “package” along the review could be beneficial.

Minor typo/editing concerns:

  • Pannexins are not hemichannels, they are channels. See line 76. This is an incorrect description of Panx channels.
  • According to your abbreviation definition, HC is already plural from “hemichannels”. Do not use HC’s. This is incorrect. Same goes for GJC’s.
  • Watch for hemi-channel/hemichannel, chose one and be consistent.
  • Lots of filler words like “however”, “moreover”, “therefore”, “most important”
  • Avoid the use of “it”, “this”, “these” as sentence subject. Avoid starting sentences with “it is important…”

Author Response

Thank you very much for your time and patience to read our manuscript and your comments and suggestions.
As you can imagine we put lot of effort to prepare this review. As experimental arrhythmolo-gists investigating the implication of gap junction connexin channels in development of cardiac arrhythmias and possibilities to prevent them we were encouraged by findings dealing with the function of Cx43 hemichannels and pannexin channels. We believe that the activation of these channels is most likely the first step facilitating pro-arrhythmia conditions. Inflammation and oxidative stress accompanies diseases, like hypertension, diabetes, obesity, etc., or acute cardiac injury. Thus, the implication of cardiomyocyte and non-myocyte Cx43 hemichannels and pannexin channels in adverse signalling seems highly relevant in these conditions. Activation of these channels deteriorates gap junction and ion channels function and facilitate pro-fibrotic processes. This topic is indeed attractive to search. Therefore, the purpose of our review was to highlight this issue for further research because of promising way to reveal novel approaches in treatment of life-threatening arrhythmias.
Comments and Suggestions for Authors
Major concerns
This review needs extensive editing work from the organization and English language point of views. For example, sub-paragraphs within each chapter need to be more focused and separated from each other’s. It reads more like a mismatch of information. There is a lack of flow, transitions, and logical progression.
We are very sorry for English language handicap despite the article was edited by English native speaker experienced in scientific reports editing. We did our best when preparing this manuscript but we agree with you that the topic of this review would be much better presented by more skilful researcher. On the other hand, this review (for the first time) focused on highly relevant role of connexin hemichannels and pannexin channels in arrhythmogenesis. Moreover, this review points out (for the first time) the mutual interaction (crosstalk) between gap junction channels and hemichannels/ panexin channels present in caveolaes and mitochondria, as indicated by topography of these entities in EM images. We believe that these facts could be attractive for the readers and challenging further investigation. In this context, we would appreciate your support despite your criticisms.
Revised manuscript includes new section 1 and we changed the order as well as the titles of some Sections.Recent studies of Panx1 structure by cryoEM revealed a heptameric assembly, not hexameric (Nature Structural & Molecular Biology volume 27, pages373–381(2020); eLife 2020;9:e54670; Nature volume 584, pages646–651(2020)). Please correct line 459.
Thank you for the newest information, which we included in the manuscript, as highlighted by blue colour.
Chapter titles do not seem to always match their content or there seems to describe overlapping topics. For example, inflammation or fibrosis formation due to activation of the purinergic receptor through release of ATP is repeated in Chapter 2 and 3.
We agree with you that it seems that Chapter 2 is similar to Chapter 3. However, Chapter 2 provides general information dealing with HC and Panx1 channel function observed mostly in non-cardiac tissues. While Chapter 3 provides evidence about HC and Panx1 channel function observed dominantly in the heart tissues and in close association to arrhythmogenesis. Therefore, we would like to keep it both if you would agree.
Nevertheless, in revised version the title of Chapter 2 (now is 3) was changed for more appropriate: Implication of HC and Panx1 channels in disturbances of ionic and redox homeostasis as well as pro-inflammatory and pro-fibrotic signaling. The title of Chapter 3 (now is 5) was changed for more appropriate: Impact of HC and Panx1 channels activity on development of cardiac arrhythmias
Figure 3 legend is misleading. Looks like a representation of the channels permeability rather than “illustrating possible implication of putative mitochondrial HC and Panx1 channels in cardioprotective signaling pathways”. May need to be revised to clearly convey your message.
Yes, we agree with you and we did correction.
Too many references seem to be reviews. References to actual research papers should be preferred.
We did effort to include the most of relevant and the most recent references, as it was also recommended by another reviewer. According to the number it is clear that it was indeed a hard work.
Cx hemichannels and Pannexin channels do not have the exact same function and regulation. A note on their differences and not considering them as the same “package” along the review could be beneficial.
Yes, we agree with you and we included appropriate information about as highlighted by blue (lines 132 and particularly 176…)
Minor typo/editing concerns:
Pannexins are not hemichannels, they are channels. See line 76. This is an incorrect description of Panx channels.
According to your abbreviation definition, HC is already plural from “hemichannels”. Do not use HC’s. This is incorrect. Same goes for GJC’s.
Watch for hemi-channel/hemichannel, chose one and be consistent.
Lots of filler words like “however”, “moreover”, “therefore”, “most important”
Avoid the use of “it”, “this”, “these” as sentence subject. Avoid starting sentences with “it is important…”
Thank you, we hope it was eliminated in revised version.

Round 2

Reviewer 1 Report

Dear Authors,   Thank you for your responses and a paper with highlighted changes.   I reviewed the paper once again and I did not notice (after reading the Author's response to my comments) any significant improvements made to the paper according to my suggestions (e.g. there is no Table summarizing the current data on the topic) I think that the changes made were introduced according to other reviewers suggestions.   I think that the value of the paper could be significantly improved by introduction of suggested changes - the paper should be more clinically oriented - the theses on potential clinical utility should be based on performed research.

Author Response

Dear reviewer,

we would like to thank you for your time to read our manuscript and we appreciate very much your critical clinical view. Unfortunately, we are very sorry that because the topic of this review is based on experimental data that challenge further investigation we could not response to your suggestions. However, we believe that this review is indeed challenging for further basic research as well as preclinical studies that may provide a new information relevant for treatment malignant arrhythmias in clinic. 

I read article entitled ‘Cardiac connexin-43 hemichannels and pannexin1 channels: provocative antiarrhythmic targets' with great interest.

This review paper concerns important topic: potential clinical utility of connexin-43 hemichannels and pannexin1 channels in cardiac arrhythmogenesis.

After initial excitement on the topic as a clinician-scientist I did not find any useful and easy to get clinical knowledge from this paper. There is insufficient data claiming the clinical utility of these potential drug targets.

You are right that there is insufficient data supporting the idea that Cx43 hemichannels and pannexin channels might be an antiarrhythmic target. Therefore, we put in the title questioner. However, as presented in Section 3 and particularly in Section 5 in revised manuscript, it is clear that activation of these channels is pro-arrhythmic. In particular, their activity during inflammation and oxidative stress, which accompanies heart diseases or myocardial injury (in patients as well). To understand the link between cardiac arrhythmias and hemichannels we included new chapter 1.

Few tables relating these potential targets to data from human studies (including those on the presence of arrhythmia – clinical studies) would improve the paper.

Unfortunately, clinical studies are missing but we are optimistic for future. See please most recent editorial in European Heart Journal, 2020, 41, 4100–4102: The sooner, the better: anti-inflammation in acute myocardial infarction as well as associated paper of Bouabdallaoui N. in this issue.

Why the authors think that these (hemi-)channels are more promising than other myocardial channels?

Maybe because we are focusing on hemichannels in this review it makes impression that we eliminated another ions channels. However, we included them as well (see f.e. Na+,K+ and Ca+ channels in the text). We also noted that activation of hemichannels can impairs function of these selective ion channels.

How currently used antiarrhythmic drugs affects these (hemi-)channels?

This is very good question but no answer yet. However, we think that not only antiarrhythmic drugs (e.g. beta blockers) but also cardioprotective drugs (e.g. statins, ARB) may influence (inhibit) hemichannels. It should be, however, proved.

Moreover, the methodology of such a paper should be a systematic review and be performed in accordance to appropriate international guidelines (e.g. PRISMA guidelines).

We used PubMed data base and included most relevant and most recent studies as it is generally used when preparing basic research review.

Reviewer 3 Report

The reviewer really appreciates the author’s efforts to respond to his comments.

The new introduction section that focuses on cardiac arrhythmias and its development was really appreciated. The reviewer especially liked the last paragraph (line 82-87); this nicely outlined the review and made the goals more clear and attractive. The separation between the GJC (section 4) and HC/Pan (section 5) was also helpful.

Minor editing comments:

Line 18, 25, 493: the reviewer still found some “GJC’s” and “HC’s”. Do not use ‘s.

Line 86: remove “is” in “is aims”

Line 165: “cardiomyocyte membrane gap junction”, did you mean Intercalated discs?

Line 191 & 372: revise the sentences starting with “while”. The current sentence structures are incorrect. “While….,…”

Line 239: “appears “as”” is missing

Line 285: Pannex1 instead of Panx1

Line 344: affect instead of effect

Line 396: consequently

Author Response

Dear reviewer,

THANK YOU SO MUCH FOR YOUR KIND ASSISTANCE AND SUPPORT.

We apologize very much for typing mistakes that we eliminated according to your comments.

The new introduction section that focuses on cardiac arrhythmias and its development was really appreciated. The reviewer especially liked the last paragraph (line 82-87); this nicely outlined the review and made the goals more clear and attractive. The separation between the GJC (section 4) and HC/Pan (section 5) was also helpful.

Minor editing comments:

Line 18, 25, 493: the reviewer still found some “GJC’s” and “HC’s”. Do not use ‘s.

Line 86: remove “is” in “is aims”

Line 165: “cardiomyocyte membrane gap junction”, did you mean Intercalated discs?

Line 191 & 372: revise the sentences starting with “while”. The current sentence structures are incorrect. “While….,…”

Line 239: “appears “as”” is missing

Line 285: Pannex1 instead of Panx1

Line 344: affect instead of effect

Line 396: consequently
